# Acute Effects of Metformin and Vildagliptin after a Lipid-Rich Meal on Postprandial Microvascular Reactivity in Patients with Type 2 Diabetes and Obesity: A Randomized Trial

**DOI:** 10.3390/jcm9103228

**Published:** 2020-10-09

**Authors:** Alessandra Schiappacassa, Priscila A. Maranhão, Maria das Graças Coelho de Souza, Diogo G. Panazzolo, José Firmino Nogueira Neto, Eliete Bouskela, Luiz Guilherme Kraemer-Aguiar

**Affiliations:** 1Post-Graduate Program in Clinical and Experimental Physiopathology (FISCLINEX), Faculty of Medical Sciences, State University of Rio de Janeiro, Rio de Janeiro, RJ 20550-013, Brazil; a_schiappacassa@hotmail.com (A.S.); priscilamaranhao@gmail.com (P.A.M.); mgcsouza@gmail.com (M.d.G.C.d.S.); drdiogopanazzolo@gmail.com (D.G.P.); 2Laboratory of Clinical and Experimental Research on Vascular Biology (BioVasc), Biomedical Center, State University of Rio de Janeiro, Rio de Janeiro, RJ 20550-013, Brazil; eliete.bouskela@gmail.com; 3Lipids Laboratory (Lablip), Policlínica Piquet Carneiro, State University of Rio de Janeiro, Rio de Janeiro, RJ 20550-003, Brazil; firminouerj@gmail.com; 4Obesity Unit, Policlínica Piquet Carneiro, Department of Internal Medicine, Faculty of Medical Sciences, State University of Rio de Janeiro, Rio de Janeiro, RJ 20550-030, Brazil

**Keywords:** diabetes, microcirculation, metformin, vildagliptin, postprandial

## Abstract

Background: Type 2 diabetes mellitus and obesity are both related to endothelial dysfunction. Postprandial lipemia is a cardiovascular risk. Notably, it is known that a high-fat diet may elicit microvascular dysfunction, even in healthy subjects. Since anti-diabetic drugs have different mechanisms of action and also distinct vascular benefits, we aimed to compare the results of two anti-diabetic drugs after the intake of a lipid-rich meal on microcirculation in patients with type 2 diabetes and obesity. In parallel, we also investigated the metabolic profile, oxidative stress, inflammation, plasma viscosity, and some gastrointestinal peptides. Subjects/Methods: We included 38 drug-naïve patients, all women aged between 19 and 50 years, with BMI ≥ 30 kg/m^2^. We performed endothelial measurements and collected samples before (fasting) and after the intake of a lipid-rich meal at 30, 60, 120, and 180 min. Patients were randomized to metformin or vildagliptin, given orally just before the meal. Endothelial function was assessed by videocapillaroscopy and laser-Doppler flowmetry to investigate microvascular reactivity. Besides, we also investigated plasma viscosity, inflammatory and oxidative stress biomarkers, gastrointestinal peptides, and metabolic profile in all time points. Results: No differences at baseline were noted between groups. Vildagliptin increased glucagon-like peptide-1 compared to metformin. Paired comparisons showed that, during the postprandial period, vildagliptin significantly changed levels of insulin and glucagon-like peptide-1, and also the dipeptidyl peptidase-4 activity, while metformin had effects on plasma glucose solely. Metformin use during the test meal promoted an increase in functional capillary density, while vildagliptin kept non-nutritive microvascular blood flow and vasomotion unchanged. Conclusions: After the intake of a lipid-rich meal, the use of vildagliptin preserved postprandial non-nutritive microflow and vasomotion, while metformin increased capillary recruitment, suggesting protective and different mechanisms of action on microcirculation.

## 1. Introduction

The prevalence of obesity has continuously increased over the past decades. In parallel, cardiovascular diseases and type 2 diabetes have also presented this increment [1]. Endothelial dysfunction occurs at the early stages of atherosclerosis and is associated with obesity independent of hyperglycemia [2]. Under physiological conditions, microcirculation is responsible for regulating tissue perfusion, optimizing oxygen diffusion, nutrient delivery, and cell excreta, with marked impairments in diabetes [3]. Many mechanisms are involved in this process, including insulin resistance, inflammation, and oxidative stress. Overexpression of endothelin-1 and reduced bioavailability of endothelium-derived nitric oxide are both related to hyperglycemia, but also to hyperinsulinemia. These phenomena lead to impaired microvascular flow velocity, density, and, finally, impaired tissue perfusion.

Although humans spend more of their day in postprandial periods, clinical studies have investigated microvascular reactivity predominantly during fasting [4]. There is a consensus that many chronic health problems, first noted in Western countries but that have progressively flourished worldwide, relate mainly to dietary habits. Mostly, modern dieting is typically composed of high-calorie meals, consisting primarily of carbohydrates and fat. However, a lipid-rich diet may cause acute damage/dysfunction to vascular function, which suggests that atherosclerosis would also occur during postprandial periods [5]. Previously, we have demonstrated that the intake of a lipid-rich meal induces postprandial microvascular dysfunction in healthy subjects and patients with obesity [6].

Metformin, a first-choice drug for diabetes, proved beneficial effects on vascular function, oxidative stress, and low-grade inflammation [7]. Dipeptidyl peptidase-4 (DPP4) inhibitors are widely used for diabetes. They have incretin mimetic and insulinotropic actions [8], and vildagliptin belongs to this class of drugs. Metformin and vildagliptin have complementary effects on the treatment of diabetes. They are usually concomitantly prescribed.

There is a lack of studies aimed at testing the effects of anti-diabetic drugs on postprandial microvascular reactivity. Possibly, some beneficial effects could be related to inflammation, oxidative stress, or even incretin mimetic effects. We aimed to compare the effects of vildagliptin and metformin on postprandial microvascular function in patients with type 2 diabetes and obesity. We hypothesized that the regulatory mechanisms of microvascular actions of these drugs during the postprandial period would be related to different mechanistic pathways.

## 2. Subjects and Methods

This study is a short-term randomized trial testing vildagliptin and metformin during the postprandial period in recently diagnosed type 2 diabetic drug-naïve patients with obesity. To minimize sex differences in vascular reactivity, we recruited only women. All patients were being followed in the Obesity Unit of the Policlínica Piquet Carneiro at the State University of Rio de Janeiro, a multidisciplinary outpatient care unit. This trial was conducted according to guidelines set out in the Declaration of Helsinki, approved by the local Research Ethical Committee (COEP: 0.87.3.2012), and registered in clinicaltrials.gov (NCT01827280).

During the first visit, the study was explained, and all patients gave us written informed consent. A clinical exam was performed, aiming to evaluate the inclusion and exclusion criteria. We selected only women aged 19–50 years old, with body mass index (BMI) ≥ 30 kg/m^2^, and an abdominal circumference ≥ 80 cm. We also asked about concomitant drugs in use. Besides the inclusion criteria mentioned above, these patients should be drug-näive, which means they were not allowed to be on the use of any drugs for the treatment of hyperglycemia. Blood samples were collected to determine whether they had or did not have the diagnosis of type 2 diabetes. They underwent a 75-g oral anhydrous glucose tolerance test (fasting and 2 h). Type 2 diabetes was diagnosed with a fasting plasma glucose (PG) ≥ 126 mg/dl and or post-load PG ≥ 200 mg/dl [9]. Those that were asymptomatic repeated PG and the diagnosis was made after another fasting PG ≥ 126 mg/dl.

Exclusion criteria were: uncontrolled hypertension, pregnancy, significant illnesses, previous myocardial infarction or angina pectoris, postmenopausal status, hematological diseases, triglycerides ≥ 400 mg/dl, smoking, known diagnosis of diabetes on use of antihyperglycemic drugs, and also aspirin, hormonal contraceptives, anticoagulants or statins, and a significant weight loss (≥5%) during the last six months. During the trial, no change in antihypertensive drugs or dosages was allowed.

We interviewed and selected 246 women. Forty patients met all criteria and were included. After inclusion, two patients (one in each group) were excluded due to: non-compliance (*n* = 1) and significant weight loss (*n* = 1) before the first test meal.

### 2.1. Study Design

After fulfilling the eligibility criteria, another visit was appointed to assess microvascular reactivity at the Laboratory of Clinical and Experimental Research on Vascular Biology (BioVasc) at the Biomedical Center on the University Campus. We advised patients not to exercise on the day before the test-meal and also not eat anything far from their habitual dieting. Nutritive and non-nutritive microvascular reactivity was tested at the following time points: fasting, and 30, 60, 120, and 180 min after the intake of a lipid-rich meal without the ingestion of any drug. Then, patients were randomized 1:1 to use metformin (1700 mg/day) or vildagliptin (100 mg/day) for 30 days. A random numerical sequence was electronically built and resulted in an aleatory numerical sequence. Independently, it always considered the final proportion of 1:1 of patients receiving metformin or vildagliptin. Anyone involved in the study participated in randomization, which was performed by an outer member from the trial.

After 30 days of using the pills, another appointment was scheduled to perform the second test. This test followed precisely the same procedure as the first one, but according to the patient’s group, a pill of metformin (850 mg) or vildagliptin (50 mg) was ingested just before meal intake.

We opted for this design to minimize the antihyperglycemic effects of the drugs tested since all patients had a recent diagnosis of diabetes and were all drug-naïve. Accordingly, while the first test was performed without any medication, the second was on the use of metformin or vildagliptin. Since we have diagnosed diabetes, we chose to keep patients on drug use before the second test. 

### 2.2. Anthropometric, Clinical, and Laboratory Measurements

The same trained examiner collected anthropometric measurements in duplicate: the waist at its smallest point with the abdomen relaxed and weight using a digital scale (Filizola, São Paulo, SP, Brazil). BMI is the weight (kg) divided by the square of height (m). Blood pressure was measured twice in the supine position with a 5-min interval (multiparameter patient monitor, Lifewindow LW6000; Digicare Biomedical Technology, West Palm Beach, FL, USA).

### 2.3. Lipid-Rich Meal

This meal consisted of foods rich in total fat, especially saturated fat, based on whole milk with chocolate, bread croissant-type with butter, cheddar cheese, and salami, with the following amounts: whole milk (200 mL), chocolate (10 g), margarine (20 g), croissant (1 unit), cheddar cheese (60 g), and salami (31 g). This meal had 564.6 kcal, with 28.9%, 20.7%, and 69.7% of carbohydrates, proteins, and total fat. Saturated fat represented 21.9 g of the total fat. We have previously validated this meal as a challenge to microvascular function, even in healthy subjects [6].

### 2.4. Microvascular Reactivity and Vasomotion

After a 12-h fast, patients were accommodated in a room with a temperature at 22° ± 1°C for 30 min before the beginning of the test. They were comfortably seated in a chair with the left arm put at the heart level and had the arm blood pressure measurement carried out in this position. The forearm and hand (dorsal part up) were put to rest on a cushion, and the 3rd finger was placed on a pedestal-mounted in the X-Y status of a Leica DM/LM microscope (Leica, Wetzlar, Germany). The top of this finger was gently immobilized by a clip to avoid minimum movements. Experiments were carried out with the continuous recording of temperature from the finger skin.

Videocapillaroscopy at baseline and after 4 min ischemia assessed the nutritive skin microcirculation. It was performed at the dorsum of the third finger and analyzed according to our standardized well-validated methodology previously described [10]. The same observer performed and read the exam, but was blinded to the patient’s group. Functional capillary density (FCD) at baseline and during post-occlusive reactive hyperemia (FCD during PORH) was assessed, which, respectively, represent the number of capillaries/mm^2^ with blood flow at baseline and after ischemia with 250× magnification. We employed this technique five times (before (fasting) and at 30, 60, 120, and 180 min after lipid-rich meal intake).

We also assessed non-nutritive skin microflow and microvascular vasomotion by laser-Doppler flowmetry (LDF; Perimed AB, Stockholm, Sweden). These measures were continuously made by a probe positioned at the dorsal side of the left wrist. Absolute amplitude variations within each frequency band were analyzed, as well as their normalized values, which are the interval of mean amplitude divided by the total spectrum mean amplitude [11]. We used Perisoft software (PSW version 2.50, Perimed AB, Stockholm, Sweden) for data analysis and fast Fourier transform to determine the contribution of different vasomotion frequency components to signal variability. We have assumed five frequency intervals, as previously defined, in the spectrum between 0.01 and 1.6 Hz: (a) endothelial (0.01–0.02 Hz); (b) neurogenic (0.02–0.06 Hz); (c) myogenic (0.06–0.15 Hz); (d) respiratory (0.15–0.4 Hz); and (e) cardiogenic (0.4–1.6 Hz) activities [12].

### 2.5. Plasma Viscosity

Plasma viscosity was assessed at all-time points above described and according to a previously validated protocol [13]. Immediately after blood collection, 5 mL of blood was centrifuged for 5 min at 1500/g. The supernatant was collected, and 0.5 mL was used to test plasma viscosity (ηp), which was assessed with a cone-in-plate viscosimeter (DV-II+ PRODigital, Brookfield Engineering Labs, Middleboro, MA, USA) at 230 s^−1^ shear rates at 37 °C. Results were expressed as mPascal × sec (mPa.s).

### 2.6. Biomarkers of Low-Grade Inflammation, Oxidative Stress, and Gastrointestinal Peptides

All assays were performed according to manufacturer protocols in the BioVasc or in the Lipids Laboratory at Biomedical Center. Plasma levels endothelin-1 were evaluated by Quantikine^®^Endothelin-1 ELISA kits (R&D Systems, Minneapolis, MN, USA). For the determination of serum levels of oxidized low-density lipoprotein (LDL), we used the Mercodia ELISA kit (Mercodia, Uppsala, Sweden). The intra- and interassay precisions were <10%.

Active glucagon like peptide-1 (GLP-1) and glucose-dependent insulinotropic peptide (GIP) were quantified by sandwich high-sensitivity ELISA chemiluminescent assay (Merck-Millipore, Billerica, MO, USA). Multiplexing analysis determined levels of insulin and glucagon by Magnetic Milliplex^®^ MAP kits (EMD Millipore, Billerica, MA, USA). All intra- and interassay precisions were <10% and <20%, respectively.

DPP4 activity was determined by glycyl-prolyl-paranitroanilide (Gly-Pro-pNA, Sigma-Aldrich, Saint Louis, MO, USA) as a chromogenic substrate. After an incubation period, the activity of DPP4 was determined by comparing the optical density of each sample with the optical density derived from the *p*-nitroaniline standard curve generated by data analysis software (KC Junior, Bio Tek, Winooski, VT, USA). Results are expressed as μM of *p*-nitroaniline/mL/min. The sensitivity of this method was 0.1 μM/mL/min, and the intra-assay precision was <3%.

Ultrasensitive C-reactive protein (CRP) was tested by the automated analyzer A25 BioSystems^®^ (Biosystems SA, Barcelona, Spain) and was measured by turbidimetry using the high-sensitivity latex method. PG, total cholesterol (TC), high-density lipoprotein cholesterol (HDL-c), and triglycerides (TG) were automated evaluated (A25 BioSystems^®^, Biosystems SA, Barcelona, Spain) spectrophotometrically by glucose oxidase/peroxidase, cholesterol oxidase/peroxidase, direct detergent, and glycerol 3-phosphate/peroxidase, respectively. LDLc was calculated using the Friedewald equation [14].

### 2.7. Statistics

We used GraphPad Prism^®^ 5 (GraphPad Software Inc., San Diego, CA, USA) for statistical analysis. The Gaussian distribution was checked. Data are expressed as mean ± SD. We employed the unpaired *t*-test and *U* test according to data distribution to compare groups. The behavior of gut peptides and metabolic profile during postprandial states are distinct from what we expect to occur in microvascular tissue, since periods of intervals between one measure to the other let microcirculation return to its resting state. Therefore, to optimize and standardize data obtained during fasting plus the postprandial period, we performed individual regression equations for modeling the relationship between independent (time) and dependent variables (e.g., components of microvascular function). A regression line was obtained, and the intercept was the point where it crosses the axis of the dependent variable. Intercepts were compared between groups. We used the G*Power 3.1.9.2 (Universit ät Kiel, Germany) to calculate the sample size of 17 patients/group. We expected a dropout rate of 20% and a total of 40 patients (FCD during PORH for group 1 and 2, respectively, of 22.8 and 32.6 cap/mm^2^; SD within each of 8.0 cap/mm^2^ according to Buss el al. [11]; effect size of 1.251; α probability error of 0.05; and a power of 0.95). A *p*-value < 0.05 was considered statistically significant.

## 3. Results

Thirty-eight patients aged 39.4 ± 6.5 years with BMI 37.3 ± 5.0 kg/m^2^ completed the study. At baseline, groups were the same concerning clinic-anthropometric and laboratory data (Table 1). They also had the same plasma viscosity, respectively, for vildagliptin and metformin groups (1.9 ± 0.1 vs. 1.8 ± 0.1 at 30 mPa.s, *p* = 0.36, and 1.8 ± 0.1 vs. 1.8 ± 0.1 at 50 mPa.s, *p* = 0.41). Additionally, considering the intercepts derived from the regression lines obtained during the test meal, groups did not have any significant differences between them at baseline (Table 2). This comparison between groups at the second test meal demonstrated that GLP-1_intercept_ was different, possibly due to changes in the vildagliptin group (Table 2). Besides this difference in GLP-1_intercept_ in paired comparison in the vildagliptin group, insulin_intercept_ and DPP4 activity_intercept_ also changed. Vildagliptin use did not alter PG_intercept_, while metformin influenced only PG_intercept_ results without any other paired effects (Table 2). No significant differences in plasma viscosity_intercept_ were noted.

Results on microvascular reactivity are depicted in Table 3 and Table 4. Concerning comparisons between groups on nutritive microvascular reactivity at baseline, any difference was observed, but we noticed that metformin showed higher FCD_intercept_ during PORH compared to the vildagliptin group (Table 3). This finding points to beneficial aspects of metformin related to capillary recruitment during PORH (nutritive microflow) even after the intake of a lipid-rich meal. We also performed a paired analysis, but no significant changes were observed.

Data on non-nutritive microflow and vasomotion during the test meal are depicted in Table 4. Baseline comparisons between groups did not show any significant differences. Still, at the second test meal, vildagliptin showed higher values of microflow during almost all of the tested period (from 30 to 180 min), while it also promoted a higher total frequency interval at the end of the postprandial period (180 min). These findings point to the beneficial effects of vildagliptin on non-nutritive microflow and vasomotion. Considering each of the five components of vasomotion separately, we did not notice any differences. We additionally investigated paired changes and observed no difference in the vildagliptin group concerning vasomotion. In contrast, metformin changed it during the postprandial state, resulting in lower mean values of vasomotion (*p* < 0.05), which was more relevant at the end of the postprandial period, as follows: 30 min (11.4[7.8–14.2] vs. 10.2[5.7–12.3] Hz; *p* = 0.03), 60 min (11.3[8.1–15.8] vs. 10.1[5.4–13.2] Hz; *p* = 0.04), and 120 min (10.7[7.4–16.6] vs. 10[5.5–11] Hz; *p* = 0.04).

## 4. Discussion

Although the benefits promoted by metformin and DPP4 inhibitors on controlling diabetes are well-defined, many investigators have also shown the effects of these drugs beyond glycemia [15,16], especially on endothelial function. Until now, no study compared their impact during the postprandial period after a lipid-rich meal. In this study, we investigated it and observed that both drugs have some benefits. Metformin promoted a marked impact on postprandial nutritive microflow, i.e., capillary recruitment. Vildagliptin showed benefits on postprandial non-nutritive microflow and vasomotion. To investigate possible mechanistic pathways, we also assessed many biomarkers related to metabolic profile, gastrointestinal peptides, inflammation, and oxidative stress.

In healthy individuals, the physiological increase of postprandial insulin levels reduces the endogenous very low-density cholesterol and inhibits lipolysis through activation of the lipoprotein-lipase. After the intake of a lipid-rich meal, patients with obesity and diabetes present an exaggerated and prolonged elevation in triglycerides and inflammatory markers [17], reaching a peak after 4 h that lasts for up to 8 h. This phenomenon may predispose them to alterations on lipid metabolism with a trend towards the development of atherosclerosis [18] named as dietary lipotoxicity. Endothelial cells may be particularly susceptible to this damage [19]. Insulin in its physiological role acts on vascular tissue promoting increments on blood flow and tissue perfusion, expressed by increased capillary recruitment. In a previous study, we observed the opposite. We compared patients with obesity and controls and noticed that a lipid-rich meal might cause postprandial microvascular dysfunction, even in lean controls [6].

Dipeptidyl peptidase-4 is expressed on endothelial cells, especially in the microcirculation [20]. Its activity and expression are increased in vitro by high glucose only in microvascular endothelial cells [21]. This phenomenon provides a rationale for the inhibition of DPP4 to protect the microcirculation from hyperglycemia. In a recent study, we noted that DPP4 activity was associated with early markers of endothelial proinflammatory activation and microvascular function [22]. Besides, other data also suggested that DPP4 inhibitors may benefit not only the micro but also the macrocirculation [23]. Possible benefits may involve GLP-1 as a mediator, which is being viewed as an endogenous protective factor for the vascular system, especially when it peaks during a meal.

Metformin acts distinctively and possibly exerts its endothelial effects through AMP-activated protein kinase activation, which leads to phosphorylation of endothelial nitric oxide synthase resulting in vascular protection [7]. Metformin’s vascular actions also appear to the increase endothelium-derived hyperpolarizing factor and to reduce the production of derivatives from the cyclooxygenase pathway [24]. Both processes involve the improvement of endothelial function and occur predominantly on micro than macrocirculation. Therefore, it is not surprising to detect different mechanisms of action on postprandial microvascular reactivity according to the particular drug tested.

DPP4 inhibitors and metformin may also influence lipids, but results are inconsistent across trials [25,26,27]. An extensive meta-analysis has suggested that DPP4 inhibitors can lower TC and TG, probably due to effects of the GLP-1 receptor stimulation on cholesterol metabolism [28]. The incretin effect seems to reduce intestinal lymph flow, triglycerides absorption, and apolipoprotein synthesis, thus being able to limit the flow of triglycerides to circulation [29]. Studies evaluating incretin levels on the postprandial period in diabetes show distinct responses depending on fatty acids offered by the meals. We hypothesize that high levels of saturated fat that we provided to patients could have limited incretin effects on lipid metabolism since diets rich in monounsaturated fatty acids act as more potent stimulators on GLP-1 and GIP [30] than the one we have opted to test.

Although we found an increase in GLP-1 and insulin, as well as an expected reduction in DPP4 activity with vildagliptin, no significant change in plasma glucose was observed, which could not be explained only by the intrinsic properties of the drug. The extremely short-term use of vildagliptin possibly influenced the absence of these results, but this finding also reinforces the concept that this drug acted on microcirculation through other mechanisms. On the counterpart, we could not exclude the effects of metformin on plasma glucose during the postprandial period as a possible influence on the observed results on nutritive microflow and vasomotion.

Patients with diabetes and obesity need to ingest adequate amounts of fat and also to follow a dietary pattern within the nutritional recommendations. In clinical practice, it is widely known that many of them do not follow these guidelines, especially those without regular follow-up. Unfortunately, these patients may eat large amounts of fat daily. Intake of fat per meal may vary from 20 to 70 g [31], which could be associated with atherogenic lesions and be an independent cardiovascular risk factor [32]. Capillary recruitment is an important phenomenon that will improve tissue perfusion and gas/nutritive exchanges, as it is directly related to vasomotion and also to tissue nutrition and glucose metabolism [33]. Gradual deterioration of meal-related capillary recruitment was previously observed in patients with metabolic syndrome and diabetes [34]. The critical point of this finding is that this impairment in microvascular function paralleled decrements in insulin sensitivity and postprandial hyperglycemia. Therefore, possible drugs that interfere with microvascular function during the postprandial period may benefit glucose control, and this study adds knowledge to the pathophysiology of microvascular damage in diabetes.

Many biomarkers were tested without significant changes. Metformin improved glucose tolerance, and this could be related to improved microvascular function during the postprandial period, while vildagliptin’s effects after the intake of a lipid-rich meal were glucose-independent, with possible pleiotropic effects on microcirculation.

Our study limitation includes the small sample size, sex bias, and pre-test conditions. We strictly calculated the sample size concerning the primary endpoint (microvascular reactivity). The other investigated data, even those reaching statistical significance, should be viewed with caution. Microvascular changes observed after the use of both drugs were well demonstrated, but a further study with a bigger sample size is necessary to better determine the consequences of these drugs on the other tested variables. Additionally, we tested only women, and the results we found are limited to this group. Although we had advised about dieting and exercise before the test-meal day, we cannot assure they followed it, but we ensure that all patients did a 12 h fast before the tests.

In conclusion, we observed that vildagliptin and metformin were both able to improve postprandial microvascular function, probably through distinct mechanisms of action on the microcirculation.

## Figures and Tables

**Table 1 jcm-09-03228-t001:** Comparisons between vildagliptin and metformin groups on the clinic-anthropometric and laboratory variables at baseline.

	Vildagliptin (*n* = 19)	Metformin (*n* = 19)	*p*-Value
Clinic-anthropometric aspects			
Age (years)	39.0 ± 5.3	39.8 ± 7.7	0.44
Weight (kg)	94.0 ± 14.0	99.5 ± 16.1	0.31
BMI (Kg/m^2^)	36.0 ± 3.0	38.5 ± 6.1	0.25
Waist Circumference (cm)	105.6 ± 10.5	106.55 ± 12.0	0.51
Hip Circumference (cm)	116.1 ± 9.0	122.7 ± 11.9	0.09
WHR	0.91 ± 0.05	0.88 ± 0.07	0.57
Systolic BP (mmHg)	131.5 ± 20.9	134.6 ± 27.2	0.75
Diastolic BP (mmHg)	84.0 ± 13.4	83.0 ± 13.1	0.97
Mean BP (mmHg)	100.0 ± 15.6	100.2 ± 17.2	0.86
Fat mass (%)	39.5 ± 3.9	41.1 ± 4.2	0.29
Lean mass (%)	60.4 ± 6.5	59.2 ± 6.1	0.30
Laboratory aspects			
Insulin (mIU/l)	1.8 ± 0.9	1.9 ± 1.2	0.75
PG (mg/dl)	200.3 ± 95.1	190.3 ± 74.0	0.88
Post-load PG (mg/dl)	253.4 ± 60.9	255.9 ± 56.9	1.0
HbA1c (%)	8.0 ± 1.8	7.8 ± 2.0	0.64
Total Cholesterol (mg/dl)	183.9 ± 36.7	198.0 ± 37.8	0.18
Triglycerides (mg/dl)	161.0 ± 79.8	141.2 ± 78.7	0.28
HDL-cholesterol (mg/dl)	43.0 ± 8.2	46.7 ± 11.3	0.38
LDL-cholesterol (mg/dl)	108.6 ± 26.3	112.90 ± 25.9	0.13
VLDL-cholesterol (mg/dl)	32.3 ± 16.0	28.3 ± 15.7	0.29
C-Reactive Protein (mg/dl)	1.1 ± 1.4	0.9 ± 0.6	0.94
GIP (pg/mL)	24.3 ± 14.6	20.3 ± 12.3	0.32
GLP-1 (pM/l))	1.1 ± 1.0	1.0 ± 0.7	0.98
Glucagon (pg/mL)	23.5 ± 20.7	31.5 ± 53.5	0.53
LDL_ox_ (U/L)	74.8 ± 31.3	62.1 ± 16.8	0.24
Endothelin-1 (pg/mL)	1.7 ± 0.5	1.6 ± 0.5	0.83
DPP4_act_ (uM/mL/min)	9.4 ± 3.5	9.6 ± 3.7	0.86

Data are expressed as mean ± SD. BMI–body mass index; BP–blood pressure; HbA1c–glycated hemoglobin; PG–plasma glucose; TC–total cholesterol; TG–triglycerides; HDL–high-density lipoprotein; LDL–low-density lipoprotein; VLDL–very low-density lipoprotein; GIP–Glucose-dependent insulinotropic peptide; GLP-1–glucagon-like peptide-1; LDL_ox_–oxidized low-density lipoprotein; DPP4_act_–Dipeptidyl peptidase-4 activity.

**Table 2 jcm-09-03228-t002:** Effects of a lipid-rich meal with and without vildagliptin or metformin use on metabolic and inflammatory biomarkers, gastrointestinal peptides, and plasma viscosity during the postprandial period.

	Vildagliptin Baseline	Vildagliptin Post-Treatment	Metformin Baseline	Metformin Post-Treatment
PG_intercept_	226.8 ± 93.0	216.3 ± 109.2	213.8 ± 80.1	**175.3 ± 61.7 *****
Insulin_intercept_	847.8 ± 476.7	**1034 ± 636.1 *****	1001 ± 526.2	993.5 ± 415.7
Total Cholesterol_intercept_	180.3 ± 36.8	177.2 ± 40.4	188.5 ± 35.6	184.7 ± 27.6
Triglycerides_intercept_	157.4 ± 75.5	146.2 ± 80.4	135.9 ± 74.4	144.7 ± 67.6
HDL-cholesterol_intercept_	41.7 ± 8.5	42.9 ± 9.5	44.1 ± 10.9	43.5 ± 9.5
LDL-cholesterol_intercept_	107 ± 27.0	102.1 ± 22.1	116.8 ± 23.4	112.1 ± 21.7
VLDL-cholesterol_intercept_	31.0 ± 16.1	26.1 ± 8.9	27.4 ± 15.2	29.3 ± 13.6
GLP-1_intercept_	2.5 ± 1.9	**5.3 ± 4.1 ****	2.1 ± 1.1	**2.3 ± 1.7 ^&&&^**
Glucagon_intercept_	15.4 ± 28.4	26.6 ± 25.3	29.0 ± 43.6	28.1 ± 40.4
LDL_ox intercept_	44.6 ± 59.4	72.2 ± 31.0	61.5 ± 17.8	61.4 ± 21.4
Endotelin_intercept_	1.5 ± 1.3	1.6 ± 0.5	1.6 ± 0.5	1.6 ± 0.6
DPP4 act_intercept_	10.3 ± 3.4	**7.6 ± 3.0 *****	9.6 ± 3.6	9.2 ± 3.7
PlasmaViscosity 30mPa.s_intercept_	1.8 ± 0.1	1.8 ± 0.1	1.9 ± 0.1	1.9 ± 0.2
Plasma Viscosity 50mPa.s_intercept_	1.8 ± 1.1	1.8 ± 0.1	1.7 ± 0.5	1.8 ± 0.2

Data are expressed as mean ± SD of Variable_intercept_. * Paired comparisons: ** *p* < 0.01; *** *p* < 0.001. ^&^ Unpaired comparisons: ^&&&^
*p* < 0.001. Legends: BMI–body mass index; BP – blood pressure, HbA1c – glycated hemoglobin, TC–total cholesterol, TG–triglycerides; HDL–high-density lipoprotein cholesterol; LDL–low-density lipoprotein cholesterol; VLDL–very low-density lipoprotein cholesterol; GLP-1–glucagon-like peptide-1; PG–Plasma glucose; LDL_ox_–oxidized low density lipoprotein; DPP4_act_–Dipeptidyl peptidase 4 activity.

**Table 3 jcm-09-03228-t003:** Effects of vildagliptin and metformin on postprandial nutritive microvascular reactivity.

Nutritive Microvascular Reactivity.	Vildagliptin Baseline	Vildagliptin Post-Treatment	Metformin Baseline	Metformin Post-Treatment
Resting FCD_intercept_	28.4 ± 13.2	33.4 ± 12.6	36.1 ± 16.2	43.1 ± 14.4
FCD during PORH_intercept_	31.1 ± 15.9	29.1 ± 13.8	38.7 ± 19.6	**45.1 ± 14.5 ^&^**

Data are expressed as mean ± SD. ^&^ Unpaired comparison: ^&^
*p* < 0.05, FCD–functional capillary density, PORH–post-occlusive reactive hyperemic.

**Table 4 jcm-09-03228-t004:** Effects of vildagliptin and metformin on postprandial non-nutritive microvascular reactivity and vasomotion.

		TIME (min)				
	Group	Baseline	30	60	120	180
**Non-nutritive microflow**						
**Mean Value *(PU)***						
**post-treatment**	**MET**	12[5.9–15.7]	10.2[5.7–12.3]	10.1[5.4–13.2]	10[5.5–11]	10.4[5.6–16]
**post-treatment**	**VIL**	13.1[8.5–2]	12.3[7.8–18.4] ^&^	12[7.5–18] ^&^	11.4[5.7–18] ^&^	10.9[5.6–17.7]
**Mean Total *(PU)***						
**post-treatment**	**MET**	0.2[0.1–0.3]	0.1[0.1–0.3]	0.1[0.1–0.3]	0.1[0.1–0.2]	0.1[0.1–0.2]
**post-treatment**	**VIL**	0.1[0.1–0.3]	0.2[0.1–0.3]	0.1[0.1–0.3]	0.1[0.1–0.2]	0.1[0.1–0.2]
**Microvascular vasomotion**						
**Total Frequency interval *(Hz)***						
**post-treatment**	**MET**	32.4[14.2–65.3]	27.3[12.9–67.2]	25.7[13.0–66.4]	23.3[14.2–39.9]	22.2[15.2–48.6]
**post-treatment**	**VIL**	29.0[20.6–59.7]	33.1[18.6–54.3]	30.6[19.1–52.5]	30.0[5.5–48.7]	31.3[16.4–50.0] ^&^
**Endothelial *(Hz)***						
**post-treatment**	**MET**	4.2[2.2–7.7]	3.7[2.1–6.4]	3.9[2.5–8.8]	4.1[2.5–7.0]	4.4[1.8–8.6]
**post-treatment**	**VIL**	4.7[3.1–7.6]	3.9[2.2–10.9]	5.3[2.0–9.0]	4.9[2.6–8.9]	4.8[2.4–7.4]
**Sympathetic *(Hz)***						
**post-treatment**	**MET**	3.8[2.2–4.9]	3.7[2.4–6.1]	3.7[2.3–5.3]	3.73[2.39–5.6]	4.0[2.1–5.5]
**post-treatment**	**VIL**	3.7[1.8–5.3]	3.9[1.8–6.1]	3.9[1.8–9.4]	4.5[2.1–10.0]	3.9[2.1–8.4]
**Myogenic *(Hz)***						
**post-treatment**	**MET**	2.3[1.6–3.5]	2.3[1.7–3.5]	2.2[1.6–2.9]	2.3[1.6–3.7]	2.2[1.8–3.8]
**post-treatment**	**VIL**	2.6[1.7–4.1]	2.3[1.7–3.5]	2.2[1.8–3.2]	2.3[1.7–4.2]	2.1[1.7–3.8]
**Respiratory *(Hz)***						
**post-treatment**	**MET**	1.3[1.1–1.5]	1.3[1.1–1.6]	1.3[1.0–1.5]	1.3[1.0–1.5]	1.7[1.0–1.5]
**post-treatment**	**VIL**	1.2[0.9–1.4]	1.2[0.8–1.5]	1.2[0.8–1.8]	1.1[0.1–1.5]	1.2[0.9–1.5]
**Cardiogenic *(Hz)***						
**post-treatment**	**MET**	0.7[0.5–0.8]	0.7[0.6–0.8]	0.7[0.6–0.8]	0.7[0.6–0.8]	0.7[0.6–0.8]
**post-treatment**	**VIL**	0.7[0.6–0.8]	0.7[0.6–0.8]	0.7[0.5–1.0]	0.7[0.5–0.8]	0.7[0.5–0.8]

Data are expressed as medians [1st–3rd tertiles]. ^&^ Unpaired comparisons: ^&^
*p* < 0.05; PU–Perfusion units; Hz–Hertz; MET–Metformin group; VIL–Vildagliptin group.

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
