# Peer review of "Acute Effects of Metformin and Vildagliptin after a Lipid-Rich Meal on Postprandial Microvascular Reactivity in Patients with Type 2 Diabetes and Obesity: A Randomized Trial"

_jcm, 2020, doi:10.3390/jcm9103228_

Round 1

Reviewer 1 Report

This is an interesting topic. The study is generally well described and carefully performed. Microvascular function was assessed using well-recognized videocapillaroscopy and laser-Doppler flowmetry. My major concern is relatively small sample size. I suppose that presented study may have inadequate statistical power. Although preliminary, in my opinion, such data deserve.

Author Response

We wish to thank you for your comments.  In this study, we aimed to investigate the effects of vildagliptin and metformin on postprandial microvascular reactivity. We have previously studied the effects of a lipid-rich meal on postprandial microvascular reactivity (ref. 6 in the paper).

Since our primary endpoint was the effect of these drugs on microvascular reactivity, we calculated the sample size accordingly (choosing the FCD of the videocapillaroscopy as the primary outcome). In accordance, we found differences in microvascular reactivity. We do agree with you that we would need a bigger sample size to reach statistical power to all variables tested. We opted to include it as a limitation (lines 397-403).

Reviewer 2 Report

The manuscript entitled "Acute effects of metformin and vildagliptin after a lipid-rich meal on postprandial microvascular reactivity in patients with type 2 diabetes and obesity" to compare the results of two anti-diabetic drugs after the intake of a lipid-rich meal on microcirculation in patients with type 2 diabetes and obesity. However, I have IMPORTANT REMARKS on the paper:

MAJOR:

Title: Identification as a randomized trial in the title

Abstract and Introduction: Please, state the scientific reason to conduct the study. Please, add why research in women diagnosed Type 2 Diabetic obese is a strong scientific argument.

Methods:

Please add, settings and locations where the data were collected.

Include method used to generate the random allocation sequence.

Inform completely plasma viscosity was assessed.

Please describe the pre-test meal conditions. The kind of meal ingested the day before the postprandial test especially, i.e., the dinner meal, exercise, etc, can have some influence.

Please add meal contents (i.e.: fatty acid content) and background diet (by 24h recall). This is important that diets rich in monounsaturated or n-6 PUFAs tend to lower the postprandial lipid response as compared with SFAs and in the case of MUFA there is evidence of an alteration in the kinetics of the postprandial responses.

Statistics: The authors have not performed a power analysis. Based on their study design, a sample size of 17 patients/group had to be recruited, given a large effect size, an alpha-error probability of 5% and 95% power. This has not been achieved in this study. However, I am aware that adequate sample sizes in studies conducted in Type 2 Diabetic obese are hard to achieve. Therefore, I'd strongly urge the authors to be careful with statements on "statistical significance"! The authors analyzed pre-post measurements and post-treatment (30, 60, 120 and 180 min) with 10 outcome variables. This statistical analysis should not be performed using a paired or unpaired comparisons-test without additional adjustments. Rather a using two-way ANOVA with repeated measures should have been carried out, which is, by the way, robust against violations of assumptions. If the authors conclude to not use two-way ANOVA, appropriate adjustments (Sidak, Bonferroni-Holm…) must be applied for multiple testing in order to prevent alpha error accumulations!

Results:

For each group, losses and exclusions after randomization, together with reasons (i.e., CONSORT Flow Diagram)

Are normative values available from the current literature that could help interpreting the performance measures and the intervention effects of the participants?

For each primary and secondary outcome, results for each group, and the estimated effect size and its precision (such as 95% confidence interval and P value).

MINOR:

General: Use American English.

Results: Make consistent use of decimals in the results' section. In favor of readability, I'd recommend 1 decimal instead of 2 for values. The same applies to Table 1 to 4: 1 decimal is sufficient.

I suggest the authors "should" performed the area under the curve (AUC) by using paired samples t-test to examine effects of vildagliptin and metformin on postprandial non-nutritive microvascular reactivity and vasomotion.

Author Response

The manuscript entitled "Acute effects of metformin and vildagliptin after a lipid-rich meal on postprandial microvascular reactivity in patients with type 2 diabetes and obesity" to compare the results of two anti-diabetic drugs after the intake of a lipid-rich meal on microcirculation in patients with type 2 diabetes and obesity. However, I have IMPORTANT REMARKS on the paper:

We wish to thank you for your comments and suggestions.

MAJOR:

Title: Identification as a randomized trial in the title

This study is a clinical trial with a head-to-head comparison between two groups, randomized to use vildagliptin or metformin. We did agree with you and decided to include it on the title (Lines 4 and 5).

We should also emphasize that this study lacks clinical importance since we have designed it to investigate acute vascular effects during postprandial state. Therefore, the core aspiration of our work resides on mechanistic aspects of the pathophysiology of vascular dysfunction in subjects with diabetes and the consequences of these tested drugs on it. Not necessarily a clinical comparison of two drugs, aiming to define which is one would be better for a patient. With this study, we are not proposing to use one drug or the other; and that is why we opted to submit it to an issue focused on the pathophysiology of diabetes of the JCM.

Abstract and Introduction: Please, state the scientific reason to conduct the study. Please, add why research in women diagnosed Type 2 Diabetic obese is a strong scientific argument.

As described above, the scientific reason to conduct it is related to vascular tissue damage after a lipid-rich meal. It is already written in the abstract (Lines 34-35 "Notably, it is known that a high-fat diet may elicit microvascular dysfunction, even in healthy subject"). We opted to elucidate it better (Lines 35-36). Additionally, we opted to have a sentence to explain why we tested only women (Line 114). As mentioned above, we do not plan to indicate the clinical use of one drug or the other with our study. Therefore, testing only women would let us investigate better the vascular reactivity without sex difference bias on vascular function. We decided to include it as a limitation (Line 397 and Lines 403-404).

Methods:

Please add, settings and locations where the data were collected.

The settings and locations were all added (lines 114-116; Lines 142-143; Lines 216-217)

Include method used to generate the random allocation sequence.

We opted to explain it better (lines 149-152)

Inform completely plasma viscosity was assessed.

Viscosity is an easy method to perform, and we had previously validated it in a published paper of our group on Menopause Journal (ref. 13).

There is nothing else to include in the described methodology. To test plasma viscosity, we need to have a viscosimeter (already described), which must be firmly installed in a bench adapted to its assumptions (precisely the way our equipment is). We also need to collect blood, centrifuge it, and collect the supernatant plasma (also decribed). Then, we have to put the plasma in a very smooth surface (cone-in-plate) to be submitted to rotations in a temperature-controlled space (inside the equipment). By doing rotations, the equipment test the shear stress of the plasma. We have described it also, but at a technically  way.

Please describe the pre-test meal conditions. The kind of meal ingested the day before the postprandial test especially, i.e., the dinner meal, exercise, etc, can have some influence.

We advised patients not to exercise before the test day and also not to eat anything far from the habitual dieting. It was not written in the paper, but we decided to include it (Lines 143-145). We cannot assure you that they carry out all the advice. Therefore, we opted to have it as a limitation (lines 397-398 and Lines 404-406).

Please add meal contents (i.e.: fatty acid content) and background diet (by 24h recall). This is important that diets rich in monounsaturated or n-6 PUFAs tend to lower the postprandial lipid response as compared with SFAs and in the case of MUFA there is evidence of an alteration in the kinetics of the postprandial responses.

Independently of MUFA or PUFA (or anything else) on the content of the diet used on our study, both groups were randomly divided into two groups, and more importantly  they were the same before the interventions (vildagliptin or metformin). Any difference between them was noticed (table 1). Therefore, since this study is not testing the effects of a specific diet content on vascular function and instead, we are trying to determine the impact of an incretin mimetic action or a biguanide on vascular function after the same diet to two groups. We do consider meal content irrelevant to the purpose of the study

Statistics: The authors have not performed a power analysis. Based on their study design, a sample size of 17 patients/group had to be recruited, given a large effect size, an alpha-error probability of 5% and 95% power. This has not been achieved in this study. However, I am aware that adequate sample sizes in studies conducted in Type 2 Diabetic obese are hard to achieve. Therefore, I'd strongly urge the authors to be careful with statements on "statistical significance"! The authors analyzed pre-post measurements and post-treatment (30, 60, 120 and 180 min) with 10 outcome variables. This statistical analysis should not be performed using a paired or unpaired comparisons-test without additional adjustments. Rather a using two-way ANOVA with repeated measures should have been carried out, which is, by the way, robust against violations of assumptions. If the authors conclude to not use two-way ANOVA, appropriate adjustments (Sidak, Bonferroni-Holm…) must be applied for multiple testing in order to prevent alpha error accumulations!

Considering some points from your question, we have to agree with you. This statement was a worry of another reviewer. We answered him/her with the following: “In this study, we aimed to investigate the effects of vildagliptin and metformin on postprandial microvascular reactivity. We have previously studied the effects of a lipid-rich meal on postprandial microvascular reactivity (ref. 6 in the paper). Since our primary endpoint was the effect of these drugs on microvascular reactivity, we calculated the sample size accordingly (choosing the videocapillaroscopy as the main outcome). In accordance, we found differences in microvascular reactivity. We do agree with you that we would need a bigger sample size to reach statistical power to all variables tested. We opted to include it as a limitation (lines 397-403).”

In some aspects, we should agree that the best way to analyze variables after meal-intake like glucose, cholesterol, gut peptides, etc; would be by using two-way ANOVA with repeated measures. But, we opted to create a standard pattern to analyze it. We had our primary outcome as the resting FCD and the FCD at PORH (videocapillaroscopy). And, we test them at all time-points (before and after meal) but, there is an essential physiological difference between FCD and metabolic or gut peptides variables. On the intervals between 0-30 min, 30-60 min, 60-90 min, and 90-180 min, microvascular tissue returns to its resting levels. We measured resting FCD and also during PORH. The behavior of microvascular reactivity is very different from those observed in glucose and gut peptides, for example. Therefore, after lengthy discussions, we opted to establish a standard pattern of analysis, which was done by extracting the intercepts from these variables. We decided to elucidate it in the paper (line 245-248)

Results:

For each group, losses and exclusions after randomization, together with reasons (i.e., CONSORT Flow Diagram)

In lines 137-139, it was already described. We have interviewed many women (n=246). Due to inclusion and exclusion criteria, especially the one related to the use of any anti-diabetic drug, we took a long time for including 40 patients. And we had only two dropouts (one in each group) which were described. We consider that a flow diagram with such a simple description would not add necessary information to the reader. We opted to add one lacking information to the reader (line 138).

Are normative values available from the current literature that could help interpreting the performance measures and the intervention effects of the participants?

Normative values for microvascular reactivity using videocapillaroscopy and laser Doppler flowmetry are not available, and data obtained from these techniques are always used comparing them with a control group or some other comparator.

For each primary and secondary outcome, results for each group, and the estimated effect size and its precision (such as 95% confidence interval and P value).

I have answered it above, and since we had a small sample size calculated based on the primary endpoint, we have it as a limitation. It is explained (above) to you.

MINOR:

General: Use American English.

Results: Make consistent use of decimals in the results' section. In favor of readability, I'd recommend 1 decimal instead of 2 for values. The same applies to Table 1 to 4: 1 decimal is sufficient.

Thank you for your suggestion. We have made many corrections on the decimals in all tables and used only one decimal.

I suggest the authors "should" performed the area under the curve (AUC) by using paired samples t-test to examine the effects of vildagliptin and metformin on postprandial non-nutritive microvascular reactivity and vasomotion.

We also considered your suggestion before analyzing this data, but concerning the same reasons to use intercepts (above explained), we opted not to calculate the area under the curve.

Reviewer 3 Report

A well done and clear paper showing the effects of commonly used drugs on lipid-induced microcirculation and endothelial function. 

My only concern is on page 16, line 352 on the verb "believe" instead of hypothesize. Believe is more a religious than a scientific term.

Author Response

We wish to thank you for your comments. We have changed it to your suggestion (Line 366)

Round 2

Reviewer 2 Report

My last decision was rejected..